# Adipose-Derived Stem Cells and Their Derived Microvesicles Ameliorate Detrusor Overactivity Secondary to Bilateral Partial Iliac Arterial Occlusion-Induced Bladder Ischemia

**DOI:** 10.3390/ijms22137000

**Published:** 2021-06-29

**Authors:** Bing-Juin Chiang, Chun-Hou Liao, Su-Han Mao, Chiang-Ting Chien

**Affiliations:** 1College of Medicine, Fu-Jen Catholic University, New Taipei City 24205, Taiwan; bingjuinchiang@gmail.com (B.-J.C.); liaoch0321@gmail.com (C.-H.L.); orange043081@hotmail.com (S.-H.M.); 2Department of Urology, Cardinal Tien Hospital, New Taipei City 23148, Taiwan; 3Department of Life Science, College of Science, National Taiwan Normal University, Taipei 11677, Taiwan

**Keywords:** overactive bladder, detrusor overactivity, adipose-derived stem cells, microvesicles, bladder ischemia, bilateral partial iliac arterial occlusion, nerve growth factor, oxidative stress

## Abstract

(1) Background: We established a new bladder ischemia rat model through bilateral partial iliac arterial occlusion (BPAO) and investigated the therapeutic effect of adipose-derived stem cells (ADSCs) and ADSC-derived microvesicles (MVs); (2) Methods: The study included four groups: (1) sham, (2) BPAO, (3) BPAO + ADSCs, and (4) BPAO + ADSC-derived MVs. Female Wistar rats with BPAO were injected with ADSCs or ADSC-derived MVs through the femoral artery. Doppler flowmetry and real-time laser speckle contrast imaging were performed to quantify blood flow in the common iliac arteries and bladder microcirculation. A 24-h behavior study and transcystometrogram were conducted after 2 weeks. Bladder histology, immunostaining, and lipid peroxidation assay were performed. The expressions of P2X2, P2X3, M2, and M3 receptors and nerve growth factor (NGF) were evaluated; (3) Results: BPAO significantly reduced bladder microcirculation, intercontraction interval (ICI), and bladder volume and increased the amplitude of nonvoiding contraction, neutrophil infiltration, and malondialdehyde and NGF levels. ADSCs and ADSC-derived MVs significantly ameliorated these effects. The results of Western blot showed that the BPAO group exhibited the highest expression of M3 and P2X2 receptors. ADSCs significantly attenuated the expressions of M2 and P2X2 receptors. ADSC-derived MVs significantly attenuated the expressions of M3 and P2X2 receptors; (4) Conclusions: ADSCs and ADSC-derived MVs ameliorated the adverse effects of BPAO including bladder overactivity, bladder ischemia, and oxidative stress. Inflammation, muscarinic signaling, purinergic signaling, and NGF might be involved in the therapeutic mechanism.

## 1. Introduction

Epidemiological studies have demonstrated a relationship between lower urinary tract symptoms (LUTSs) and vascular risk factors including aging, obesity, diabetes, hypertension, smoking, and hyperlipidemia [1,2]. Vascular intimal thickness on carotid Doppler ultrasonography, pelvic blood flow measured through magnetic resonance imaging, and bladder neck perfusion measured through transrectal Doppler ultrasonography have all been demonstrated to predict the severity of LUTSs [3,4,5,6]. Overactive bladder (OAB) is a complex symptom that involves frequent urinary urgency and nocturia in the absence of urinary tract infection or other detectable disease. Accumulated epidemiological, clinical, and basic research evidence indicates an association of LUTSs including OAB with pelvic ischemia [7].

Azadzoi et al. first performed balloon endothelial denudation of iliac arteries and administered a high-cholesterol diet to induce pelvic arterial atherosclerosis and varying degrees of chronic bladder ischemia [8]. Ischemia-induced histopathological changes in the bladder tissue, functional changes, and detrusor instability. Subsequent animal studies on arterial endothelial denudation techniques and high-cholesterol diets have demonstrated ultrastructural changes in the bladder, oxidative stress, nerve degeneration, inflammation, and decreased nitric oxide (NO) expression [9,10,11,12,13,14,15,16]. However, a high-cholesterol diet alone may induce hyperlipidemia, detrusor overactivity, and pathological changes in the detrusor [17,18]. Increased serum low-density lipoprotein deposition was reported to enhance transforming growth factor (TGF)-β1 production and fibronectin synthesis in other organs [19]. Shenfeld et al. found that apolipoprotein E gene knockout (APOEKO) mice developed pelvic arterial atherosclerosis and chronic pelvic ischemia without functional changes in the detrusor [20]. In our previous study, we established an animal model of bilateral partial iliac arterial occlusion (BPAO) to mimic pure pelvic ischemia without confounding metabolic syndrome [21]. We found that bladder ischemia-induced substantial oxidative stress and subsequent programmed cell death. In addition, we observed that BPAO-induced detrusor overactivity (DO) with muscarinic and purinergic signaling changes. These results indicate that different degrees of bladder ischemia lead to various physiological and pathological changes in the bladder.

Within the field of urology, stem cell therapy was first applied to treat erectile dysfunction. Mesenchymal stem cells (MSCs) or adipose-derived stem cells (ADSCs) were injected into the rat corpora cavernosa to improve the functioning of the cavernosal smooth muscle with an increased NO level [22,23,24]. Three types of stem cell therapies have been used in recent studies: bone marrow-derived mesenchymal stem cells (BM-MSCs), skeletal muscle-derived mesenchymal stem cells, and ADSCs [25]. ADSCs can be easily obtained in large quantities, and their harvest and culture are associated with minimal morbidity. Previous studies have shown that ADSCs successfully treated bladder ischemia-induced using a high-fructose diet and a high-cholesterol diet [26,27]. ADSCs healed injured tissues through direct differentiation and by exerting a paracrine effect [28]. However, stem cell therapy may be compromised by tumorigenic potential [29].

Stem cells can secrete exosomes or microvesicles (MVs) that play a crucial role in signaling molecules communication between stem and recipient cells [30]. Exosomes or MVs can effectively transmit cytoprotective signals to cardiomyocytes in the setting of myocardial ischemia and reperfusion (I/R) [31]. Cumulative evidence demonstrated that ischemia, hypoxia, and I/R with the accompanying generation of reactive oxygen species are crucial etiological factors in bladder dysfunction related to bladder outlet obstruction (BOO), arterial atherosclerosis, and/or diabetes [32]. No study has used MVs for treating OAB. Thus, the current study (1) examined the effect of BPAO on physiological changes and alterations in associated motor or sensory signaling and (2) evaluated whether ADSCs and ADSC-derived MVs can effectively ameliorate BPAO-related bladder dysfunction.

## 2. Results

### 2.1. ADSC-Derived MVs Mitigated BPAO-Induced Bladder Ischemia

Our purified MVs displayed high CD29-positive expression by cytofluorimetric analysis, a spheroid morphology, and a 70 nm size with scanning electron microscopy (unpresented data). Figure 1a demonstrated the BPAO methods. Figure 1b shows changes in blood flow in the bilateral iliac arteries, as determined through Doppler flowmetry. Blood flow decreased immediately after BPAO and remained low after 2 weeks. Furthermore, 2 weeks after BPAO, blood flow in the bilateral common iliac arteries was still significantly decreased compared with that before BPAO (Figure 1c; *p* < 0.05). BPAO could reduce blood flow in the bilateral common iliac arteries to <1 mL/min.

The real-time measurements of bladder microcirculation obtained using laser speckle contrast imaging are presented in Figure 2a. A significant redistribution in the color of the bladder body was associated with a decline in high-flow areas (red) and an increase in low-flow areas in the BPAO group compared with the sham group (Figure 2b; *p* < 0.05). The BPAO + ADSC-derived MV group exhibited significant mitigation of reduced microcirculation following BPAO (Figure 2b; *p* < 0.05). In the BPAO + ADSC group, reduced microcirculation following BPAO tended to be reversed (Figure 2b).

### 2.2. ADSCs and ADSC-Derived MVs Ameliorated BPAO-Induced Bladder Physiological Changes

Figure 3a illustrates the micturition patterns determined in the metabolic cage study. The urinary frequency was significantly higher in the BPAO group than in the sham group, BPAO + ADSC group, and BPAO + ADSC-derived MV group (Figure 3b). The amount of water intake, food intake, urine output, and feces output did not significantly differ between the four groups (Figure 3c–f). The detailed data are shown in Table 1.

Figure 4a presents the IVP patterns observed through transcystometrography. A lower ICI was observed in the BPAO group than in the sham group (*p* < 0.05; Figure 4b). The ICI in the BPAO + ADSC and BPAO + ADSC-derived MV groups was higher than that in the BPAO group (*p* < 0.05; Figure 4b). The amplitude was higher in the BPAO group than in the sham group (*p* < 0.05; Figure 4c). However, the amplitude level was lower in the BPAO + ADSC-derived MV group than BPAO group (*p* < 0.05; Figure 4c). The average urine amount did not significantly differ between the four groups (Figure 4d). The residual urine amount was slightly higher in the BPAO group than in the sham group (*p* > 0.05; Figure 4e) and significantly lower in the BPAO + ADSC-derived MV group than in the BPAO group (*p* < 0.05; Figure 4e). The bladder volume was lower in the BPAO group than in the sham group (*p* < 0.05; Figure 4f). The BPAO + ADSC-derived MV group had a higher bladder volume compared with the BPAO group (*p* < 0.05; Figure 4f). The bladder weight did not significantly differ between the groups (Figure 4g). Detailed data are provided in Table 2.

Figure 5a shows micturition contraction patterns obtained through transcystometrography. The duration of phase 1 contraction and phase 2 HFOs did not significantly differ between the groups (Figure 5b,c). Detailed data are shown in Table 3.

Figure 6a presents NVCs observed through transcystometrography. The number of NVCs did not significantly differ between the groups (Figure 6b). The mean amplitude of NVCs was significantly higher in the BPAO group than in the sham group (*p* < 0.05; Figure 6c). The mean amplitude of NVCs was significantly lower in the BPAO + ADSC and BAPO + ADSC-derived MV groups than in the BPAO group (*p* < 0.05; Figure 6c). The maximum amplitude of NVCs was significantly higher in the BPAO group than in the sham group (*p* < 0.05; Figure 6d). The maximum amplitude of NVCs was significantly lower in the BAPO + ADSC-derived MV group than in the BPAO group (*p* < 0.05; Figure 6d). Detailed data are shown in Table 4.

ADSCs and ADSC-derived MVs influenced the purinergic and muscarinic cholinergic signaling caused by BAPO.

We evaluated the expression of muscarinic cholinergic proteins (M2 and M3) and purinergic receptor proteins (P2X2 and P2X3) through Western blot. Compared with the sham group, the BPAO group exhibited significantly increased expressions of M3 and P2X2 receptors (*p* < 0.05; Figure 7b,c). The expression of the M2 receptor was significantly lower in the BPAO + ADSC group than in the BAPO group (*p* < 0.05; Figure 7a). The expression of the M2 receptor also tended to be lower in the BPAO + ADSC-derived MV group than in the BAPO group; however, the difference was not significant (*p* > 0.05; Figure 7a). The expression of the M3 receptor was significantly lower in the BPAO + ADSC-derived MV group than in the BAPO group (*p* < 0.05; Figure 7b). The expression of the P2X2 receptor was significantly lower in the BPAO + ADSC and BAPO + ADSC-derived MV groups than in the BAPO group (*p* < 0.05; Figure 7c). The expression of the P2X3 receptor did not significantly differ between the groups (Figure 7d).

ADSCs and ADSC-derived MVs ameliorated inflammatory cell infiltration, increased the NGF level, and reduced the oxidative stress caused by BAPO.

Figure 8a shows the results of H&E staining for the four groups, and Figure 8b presents an analysis of neutrophil counts. The results showed that the BPAO group had the highest number of infiltrated neutrophils (*p* < 0.05). The neutrophil count was significantly lower in the BPAO + ADSC and BPAO + ADSC-derived MV groups than that in the BPAO group (*p* < 0.05). Figure 9a shows the results of NGF staining. Image analysis results showed significantly increased NGF density in the BPAO group compared with the sham, BPAO + ADSC, and BPAO + ADSC-derived MV groups (*p* < 0.05). Western blot results showed significantly increased NGF expression in the BPAO group compared with the other groups (*p* < 0.05; Figure 9b). The results of the MDA assay showed that the BPAO group had a significantly higher level of MDA than did the other groups (*p* < 0.05; Figure 10).

## 3. Discussion

The major causes of OAB are bladder ischemia and metabolic syndrome [17]. The pathophysiology of metabolic syndrome-associated OAB includes increased oxidative stress, altered regulation of postsynaptic receptors, dysregulation of smoothelin, and enhanced programmed cell death [29]. Other major etiologies are BOO and pelvic atherosclerosis-related bladder ischemia. Excessive bladder I/R injury leads to increased oxidative stress, subsequent tissue damage, and altered smooth muscle function [30]. Prolonged abnormal conditions cause the injured bladder to progress to detrusor overactivity, impaired contractility, and underactive bladder. We established a pure bladder ischemic rat model without confounding hyperlipidemia, diabetes, metabolic syndrome, or other systemic effects. We observed reduced bladder microcirculation, significant ICI shortening, and increased NVC amplitude after 2 weeks of BPAO. Urinary frequency and NVC, examined on a cystogram and by performing metabolic cage evaluation, are usually surrogate markers for OAB animal models because urgency cannot be ascertained in animals [31]. In contrast to irritative OAB models that involve the instillation of irritants in the bladder, the BPAO model causes no urothelial injuries and is therefore characteristic of not acute cystitis [31]. This phenomenon closely matches the definition of OAB. In this BPAO model, phasic contraction during the voiding phase (Figure 5) exhibited no significant interruption. Compared with the acute partial BOO model, which shows changes in the micturition reflex, an increase in phase 1 contraction, and a decrease in phase 2 HFOs, the BPAO model exhibits subtle changes in bladder function and mimics the normal aging process [32]. In addition, in the BPAO model, pelvic blood flow can be quantified and adjusted by changing the size of needles without requiring arterial endothelial denudation methods. This BAPO model mimics pelvic atherosclerotic ischemic changes in humans and can thus be used for pharmacological development.

MDA assay revealed that BPAO caused bladder ischemia and subsequent oxidative stress. Bladder ischemia caused cell infiltration; this finding is in agreement with that reported in a previous study [17]. BPAO significantly increased the expression of M3 and P2X2 receptors. Muscarinic signaling, especially the M3 receptor, is primarily responsible for detrusor contraction [33]. An altered expression level of the M3 receptor was observed in the human detrusor muscle with idiopathic DO. Overexpression of the M3 receptor has also been identified in other neurogenic bladder overactivity models [33,34]. Ischemia causes a reduction in choline acetyltransferase, resulting in the compensatory upregulation of muscarinic receptor expression and DO [35]. Purinergic signaling is responsible for the voiding reflex and occurs at multiple sites including the central nervous system, peripheral motor and sensory nerves, detrusor muscle, and urothelium [36]. Upregulation of P2X2 and P2X3 was also observed in a cyclophosphamide-induced bladder overactivity model [34]. NGF, a small signaling neuropeptide, is produced by the detrusor muscle and urothelium. NGF is necessary for the promotion and survival of the sensory and sympathetic nerves of the bladder [37]. An increased NGF level was observed in DO secondary to BOO, cyclophosphamide administration, and spinal cord injury [38]. DM-related bladder dysfunction reduces NGF production [39]. NGF is released in the inflammatory status and can modulate the expression of the P2X3 receptor. In addition, activation of the bladder C-fiber owing to various etiologies induces the release of NGF from the urothelium [40]. The findings of the current study suggested that NGF and associated purinergic signaling molecules were involved in BPAO-induced bladder overactivity.

The results of the present study indicated that ADSCs and ADSC-derived MVs exerted a therapeutic effect on ischemic bladder-related overactivity. Administration of ADSCs through femoral artery injection significantly increased ICI without affecting amplitude, bladder total volume, voiding volume, postvoiding residual urine amount, bladder weight, or phasic 1/2 contraction. In addition, ADSCs significantly normalized the mean amplitude of NVCs relative to the control group. ADSCs appeared to improve bladder microcirculation, although the results were not significant. H&E staining showed that ADSCs significantly ameliorated inflammatory cell infiltration. In addition, ADSCs reduced oxidative stress, NGF levels, and M2 and P2X2 receptor expressions. By labeling MSCs with green fluorescence, Woo et al. found that the administration of MSCs through the tail vein improved the compliance of the bladder in the rat BOO model through the chemoattractant properties of MSCs [41]. Dayanc et al. directly injected BM-MSCs into the decompensated bladder and observed reduced fibrosis and improved bladder compliance [42]. ADSCs could improve bladder function and reduce cellular apoptosis by exerting a paracrine effect on the DM rat model [22]. Lee et al. found that MSCs inhibited BOO-induced bladder fibrosis and improved DO [43]. Chen et al. reported that the injection of BM-MSCs in the common iliac artery improved detrusor muscle and nerve survivability and DO in a rat BOO model [44]. Direct injection of ADSCs into the bladder and tail vein improved muscle cell, nerve, and vessel density in rats fed with a high-cholesterol diet [23]. In addition, injection of ADSCs into the bladder prevented radiation-induced bladder dysfunction and histological changes by exerting a paracrine effect [45]. The potential therapeutic mechanism includes direct cellular differentiation, release of various growth factors, restoration of vascularity, and antioxidative properties [23]. Our study emphasized the therapeutic value of ADSCs in the treatment of OAB-induced pelvic ischemia. Future studies should elucidate the detailed mechanism.

We followed protocols described in our previous study to purify MVs from ADSCs [46]. Extracellular vesicles (EVs) include exosomes (50–150 nm) released through exocytosis, microvesicles (0.1–1 μm) shed from the plasma membrane, and apoptotic vesicles (>1 μm). The isolation protocol used in the current study resulted in the enrichment of vesicle subpopulations including exosomes and MVs [47]. These EVs possess a defined set of membrane and cytosolic proteins, messenger RNA, and microRNAs [48]. EVs play a crucial role in cellular communication and tissue regeneration. MSC-derived MVs have been used not only as a biomarker for ischemic heart disease but also to improve cardiac function or I/R injuries in cardiomyocytes [49,50,51,52,53]. The bladder and heart share similar properties in terms of chambers and repeated contraction. Pressure overload inside the chamber results in repeated I/R injury. Thus, we used ADSC-derived MVs to treat bladder ischemic. Our results indicated that the administration of ADSC-derived MVs through femoral artery injection significantly improved bladder microcirculation following BPAO. In addition, ADSC-derived MVs significantly increased bladder volume and reduced the residual urine amount and the amplitude of voiding pressure. Moreover, ADSC-derived MVs significantly normalized the mean amplitude of NVCs to the level of the control group. H&E staining showed that ADSC-derived MVs significantly ameliorated inflammatory cell infiltration following BPAO. In addition, ADSC-derived MVs reduced oxidative stress, NGF, and M3 and P2X2 receptor expression. In the field of urology, increasing attention is being paid to the use of EVs from urine as biomarkers for various diseases [54]. Although few studies have used MVs to treat kidney injuries, MVs may activate a proliferative mechanism in surviving tubular cells after injury through the horizontal transfer of mRNA [55]. The current study is the first to examine the therapeutic effects of ADSC-derived MVs on a rat model of ischemia-related OAB. The therapeutic effects of ADSC-derived MVs are comparable to those of ADSCs. Because the cell parts were removed following sequential ultracentrifugation [47], the risk of tumorigenesis of stem cells decreased. The results revealed the potential of ADSC-derived MVs in treating ischemia-related OAB.

## 4. Materials and Methods

### 4.1. Animal Groups and Experimental Design

A total of 24 adult female Wistar rats weighing 200–240 g were purchased from BioLASCO Taiwan (Taipei) and housed at the Experimental Animal Center, National Taiwan Normal University, at a constant temperature under a consistent light-dark cycle (light from 07:00 to 18:00). All animal surgical and experimental procedures were approved by the Institutional Animal Care and Use Committee of National Taiwan Normal University, College of Life Science (approval No. 106049) and performed in accordance with the guidelines of the National Science Council of the Republic of China (NSC 1997).

The rats were divided into four groups: (1) sham control (sham, *n* = 6), (2) 2 weeks of bladder ischemia-induced by BPAO (BPAO, *n* = 6), (3) ADSC injection through the femoral artery at the next week following BPAO and a total 2 weeks of BPAO-induced bladder ischemia (BPAO + ADSCs, *n* = 6), and (4) ADSC-derived MV injection through the femoral artery at the next week following BPAO and a total 2 weeks of BPAO-induced bladder ischemia (BPAO + ADSC-derived MVs, *n* = 6). The baseline of physiology parameters, namely urinary frequency, water intake, food intake, urine output (labeled as urine), and feces output (labeled as feces), were recorded and analyzed in R-2100 metabolic cages (Lab Products, Rockville, MD, USA) for 24 h in the four groups. Figure 1a illustrates the induction of BPAO. Briefly, a lower midline abdominal incision was made to expose the common iliac vessels. Subsequently, 30 G fine needles were placed beside the common iliac arteries. We ligated the common iliac arteries and fine needle together and then withdrew the needle. Then, the common iliac arteries were partially occluded. Detailed procedures are described in our previous study [17]. After BPAO, each rat received subcutaneous administration of ketoprofen (Sigma-Aldrich, Darmstadt, GER, 5 mg/kg) once daily for one week. Before, immediately after, and 2 weeks after BPAO, we used a transonic flow meter (T420, Transonic System Inc. Ithaca, NY, USA) to measure blood flow in the common iliac arteries. Blood flow in the bilateral common iliac arteries was compared between the sham and BPAO groups. Bladder microcirculation was then evaluated. After physiological experiments, the bladder was removed, and the weight was measured. One part of the bladder was fixed in 4% buffered formalin for staining. Part of the samples was stored at −70 °C for protein expression analysis through Western blot and lipid peroxidation (malondialdehyde [MDA]) assay.

### 4.2. Preparation of ADSCs and ADSC-Derived MVs

ADSC isolation was performed as described previously [56]. The flank adipose tissue was collected under sterile conditions in a centrifuge tube after administering general anesthesia in the animals. The adipose tissue was then stirred to aspirate off saline and oil phases. The fat was washed 3 to 5 times with 0.1 M phosphate-buffered saline (PBS), and the lower phase was discarded until it was clear. After collecting the final upper phase, collagenase was added; the solution was incubated for 1–4 h at 37 °C on a shaker. Subsequently, 10% fetal bovine serum (FBS) was added to the tube to neutralize the collagenase. The fluid in the tube containing digested fat was centrifuged at 800× *g* for 10 min. Then, the supernatant containing floating adipocytes, lipids, and liquid was aspirated to obtain the left stromal vascular fraction (SVF) pellet. We used 160 mM NH4Cl to suspend the SVF pellet, which was then incubated for 10 min at room temperature. The pellet was centrifuged at 400× *g* for 10 min at room temperature after incubation. The final pelleted fraction of mononuclear cells was then resuspended in Dulbecco’s modified Eagle’s medium (DMEM; Sigma-Aldrich) supplemented with 40% FBS (Invitrogen, Massachusetts, USA), penicillin–streptomycin, and 10 ng/mL of epidermal growth factor (Invitrogen) and incubated on a petri dish overnight to select adherent cells. The remaining debris and cells were aspirated off the next day, and the plate was washed with PBS. ADSCs were maintained in low-glucose DMEM supplemented with 10% FBS, 1% penicillin–streptomycin, and L-glutamine at 5% CO_2_ and 37 °C. ADSCs were maintained on a T75 flask and passaged until 80–90% confluence.

ADSC-derived MVs were isolated as described previously [57]. In brief, MVs were purified through differential ultracentrifugation. ADSCs were cultured in low-glucose DMEM without FBS and supplemented with 0.5% BSA overnight. The clear conditioned medium was transferred to centrifuge tubes and then centrifuged for 10 min at 300× *g* and 4 °C. The pellets filled with dead cells and cell debris were discarded and the supernatant was retained. The supernatant was then centrifuged three times for 10 min at 2000× *g*, 30 min at 10,000× *g*, and 30 min at 10,000× *g* at 4 °C. The final supernatant was discarded, and isolated MVs were suspended in Dulbecco’s phosphate-buffered saline (DPBS) and then recentrifuged for 30 min at 10,000× *g* and 4 °C to remove contaminating proteins. We collected the pellet containing MVs and resuspended it in DPBS. MVs were washed and ultracentrifuged at 10,000× *g* for 60 min at 4 °C. The supernatant was discarded, and MV pellets were resuspended in DPBS for later use. MVs purity was assayed with CD29 (Becton Dickinson, New Jersey, USA) and morphology was analyzed with a scanning electron microscopy (JEOL-6500F, JEOL, Japan). ADSCs or MVs were injected through the femoral artery at a dosage of 500,000 cells (an amount of cells releasing approximately 100 µg MVs overnight) or a dose of 100 μg MV proteins, respectively.

### 4.3. Measurement of Bladder Microcirculation

To determine the effects of BPAO on bladder microcirculation, a full-field laser perfusion imager (Moor FLPI, Moor Instruments, Devon, UK) was used to quantify the intensity of microcirculatory blood flow in the bladder in the four groups. This imager uses real-time laser speckle contrast continuous noncontact imaging, which exploits the random speckle pattern that is generated when the tissue is illuminated by laser light. The microcirculatory blood flow intensity in the region of interest was recorded as Flux with perfusion units. This is related to the product of the average speed and concentration of moving red blood cells in the region of interest. The contrast image produces 16-color images that are correlated with blood flow in the tissue. A high level of blood flow was recorded in red color, whereas a low level of blood flow was recorded in blue color (range, 0–1000). Images were recorded and analyzed in real-time by using Moor FLPI 3.0 software (Moor Instruments).

### 4.4. Evaluation of Transcystometrogram

Baseline micturition parameters were measured in all animals. All animals were anesthetized with subcutaneous urethane to allow full-reflex bladder contractions [58]. We used a transcystometric model to evaluate micturition parameters. In brief, rats were anesthetized by administering a subcutaneous injection of urethane (1.2 g/kg body weight). The bladder was exposed through a midline incision of the abdomen, and a PE-50 catheter was inserted through the apex of the bladder dome and connected through a T-tube to a P23 ID infusion pump and pressure transducer (Gould-Statham, Quincy, MA, USA). During the experiment, the infusion rate was maintained at 1.2 mL/h, and intravesical pressure (IVP) was continuously recorded using an ADI system (Power-Lab/16S, ADI Instruments, Pty., Ltd., Castle Hill, Australia). At the beginning of measurements, the bladder was emptied. Three reproducible micturition cycles were recorded following bladder emptying. The following parameters of bladder activity were measured: intercontraction interval (ICI), the time between two micturition cycles identified based on active contractions (>15 mmHg), average urine amount per micturition (labeled as average urine amount), bladder volume, amplitude of IVP on micturition contraction (labeled as amplitude), and residual urine amount (labeled as residual urine). In addition, we recorded and analyzed contraction phases during a micturition cycle: an initial rise in IVP (phase 1) and a series of high-frequency oscillations (HFOs; phase 2). Nonvoiding contraction (NVC) parameters, namely the number, mean amplitude, and maximum amplitude, were recorded and analyzed. NVCs were characterized by an increase in pressure of >20% from baseline, defined as the bladder pressure immediately prior to the NVC; this was observed as an increase in pressure that did not result in void [59].

### 4.5. Morphological Staining

A portion of the bladder was cut and fixed in 10% neutral buffered formalin solution, dehydrated in graded ethanol, and embedded in paraffin. Sections (4 μm) of the bladder were stained with hematoxylin and eosin to evaluate the extent of inflammatory cells. To prepare slides for immunohistochemical examination, the slides were blocked with 5% bovine serum (BioShop, Cat# ALB001; Ontario, CAN) in PBS with 0.3% Triton-X 100 (Sigma Chemical Co., Cat# X100; St. Louis, MO, USA) for 1  h. The slides were incubated overnight with primary antibodies prepared using 1% normal donkey serum and 0.1% Triton-X 100 in 0.1 M PBS at room temperature.

Nerve growth factor (NGF) was labeled using a rabbit monoclonal antibody (ab52918, Abcam) purified at a working dilution of 1:250. The secondary antibody ab97051 goat antirabbit Ig G H&L (HRP) was then applied at a dilution of 1:500. The sample was counterstained with hematoxylin. Antigen retrieval was performed using Tris-EDTA buffer (pH 9.0). PBS was used instead of the primary antibody as the negative control.

### 4.6. Western Blot and Biochemical Evaluation

The expression levels of purinergic P2X2 and P2X3 receptors, muscarinic M2 and M3 receptors, and NGF were analyzed through Western blot as described previously [60]. The bladder samples were homogenized using a prechilled mortar and pestle in an extraction buffer of 10 mM Tris-HCl (pH 7.6), 140 mM NaCl, 1 mM phenylmethylsulfonyl fluoride, 1% NP-40, 0.5% deoxycholate, 2% β-mercaptoethanol, 10 mg/mL pepstatin A, and 10 mg/mL aprotinin. The mixture was completely homogenized through vortexing and maintained at 4 °C for 30 min. The homogenate was centrifuged at 12,000× *g* for 20 min at 4 °C. The supernatant was collected, and the protein concentration was determined using the BioRad protein assay kit (BioRad Laboratories, Hercules, CA, USA). Antibodies raised against P2X2 receptors (ab10266, Cambridge, UK), P2X3 receptors (Neuromics, RA141399, Northfield, MN, USA), M2 receptors (Novus bio, nb120-2805, Novus Biologicals, LLC, Colorado, USA), M3 receptors (Abcam, ab87199, Cambridge, UK), and NGF (Abcam, ab6199, Cambridge, UK) were used. Sodium dodecyl sulfate–polyacrylamide gel electrophoresis was performed using 10% separation gels in the absence of urea and with Coomassie brilliant blue staining. For immunoblotting, proteins were transferred to Immobilon polyvinylidene difluoride membranes (Millipore, Billerica, MA, USA) for 18 h at 100 mA in a Miniprotean III transfer tank (BioRad, Hercules, CA, USA). Immunoreactive bands were detected through incubation with the aforementioned respective antibodies, followed by incubation with an alkaline phosphatase-labeled secondary antibody and western Lightning Plus-ECL stock solution (PerkinElmer, Waltham, MA, USA) for 1 min at room temperature. The density of the band was semi-quantitatively determined through densitometry by using an image analysis system (ImageJ). We determined the MDA concentration by using a lipid peroxidation assay kit (ab118970; BioVision, San Diego, CA, USA). The MDA concentration was determined according to manufacturer’s instructions.

### 4.7. Statistical Analysis

We used GraphPad Prism 6 (GraphPad Software, San Diego, CA, USA) to perform data analysis. All values are expressed as the mean ± standard error of the mean. Differences between groups were examined using a one-way analysis of variance, followed by Duncan’s multiple-range test. Furthermore, differences within groups were examined using paired *t*-tests. Differences with *p* < 0.05 were considered significant.

## 5. Conclusions

BPAO-induced bladder ischemia and subsequent oxidative stress. The associated physiological changes include bladder overactivity with a reduced bladder capacity and an increased amplitude of NVC. Inflammation, muscarinic, and purinergic receptors, and NGF might be involved in the underlying mechanism. Treatment with ADSCs and ADSC-derived MVs through femoral artery injection ameliorated adverse effects. To date, this is the first study to show the potential of ADSC-derived MVs in treating ischemia-related OAB.

## Figures and Tables

**Figure 1 ijms-22-07000-f001:**
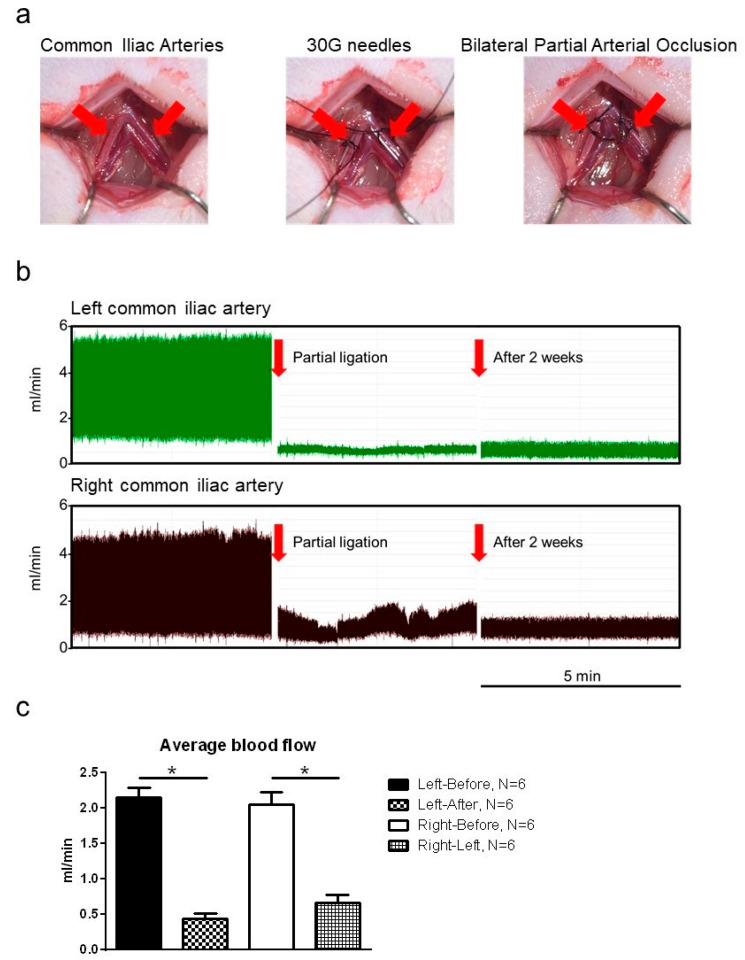
BPAO and the blood flow measurement. In the top panel, (**a**) showed the bladder ischemia model, BPAO. In the middle panel, (**b**) showed changes in blood flow of respective left and right common iliac artery with Doppler flowmetry before, after, and 2 weeks later following BPAO. In the bottom panel, (**c**) showed the blood flow of respective left and right common iliac arteries 2 weeks after BPAO significantly reduced compared to that before BPAO (* *p* < 0.05 with paired *t*-test).

**Figure 2 ijms-22-07000-f002:**
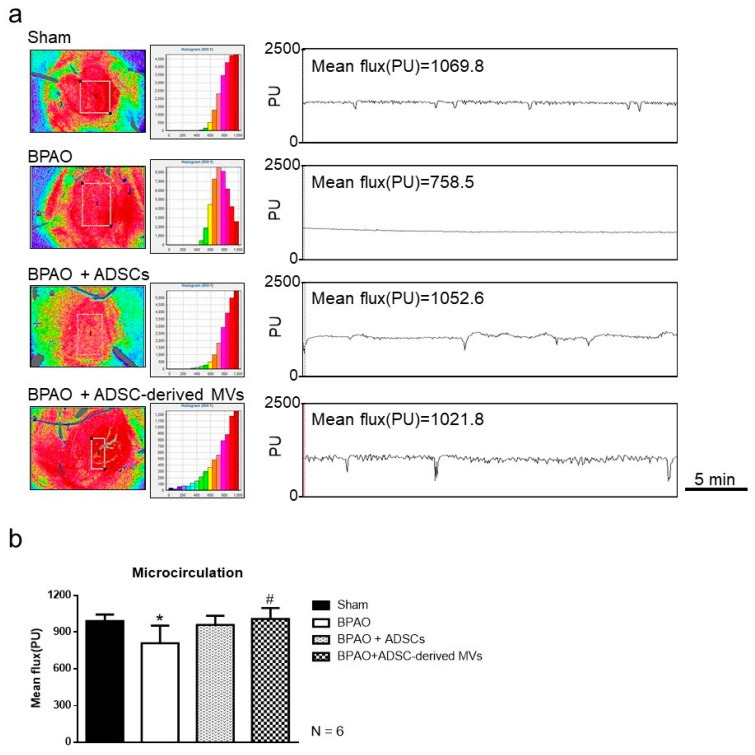
Microcirculation patterns on speckle contrast imaging study in separated group. In the top panel, (**a**) showed the real-time measurements of bladder microcirculation with laser speckle contrast imaging. In the bottom panel, (**b**) demonstrated a significant redistribution of color-scale of the bladder body associated with a decline in high-flow areas (red) and an increase in low-flow areas in BPAO group compared to sham group (* *p* < 0.05). The BPAO + ADSC-derived MVs group had significant mitigation of the reduced microcirculation following BPAO (^#^ *p* < 0.05). The BPAO + ADSCs group had trend of reversing the reduced microcirculation following BPAO (*p* > 0.05).

**Figure 3 ijms-22-07000-f003:**
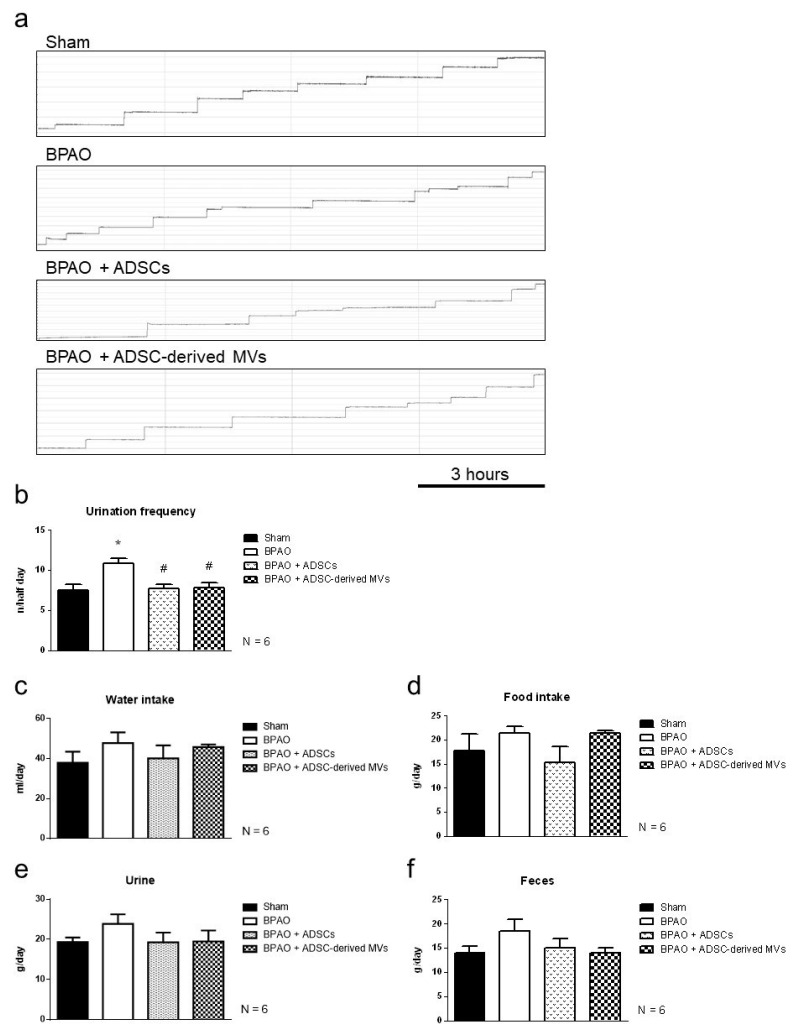
Physiology parameters on metabolic cage study in separated group. In the upper panel, (**a**) demonstrated the micturition patterns on metabolic cage study. In the lower panel, (**b**–**f**) showed statistical analysis of behavior parameters. (**b**) showed urinary frequency in BPAO group was significantly higher than sham group (* *p* < 0.05). Urinary frequency in BPAO + ADSCs, and B BPAO + ADSC-derived MVs group were significantly lower than BPAO group (^#^ *p* < 0.05). The amount of water intake, food intake, urine output, and faces output were of no significant differences among the four groups (**c**–**f**). The detailed data are shown in Table 1.

**Figure 4 ijms-22-07000-f004:**
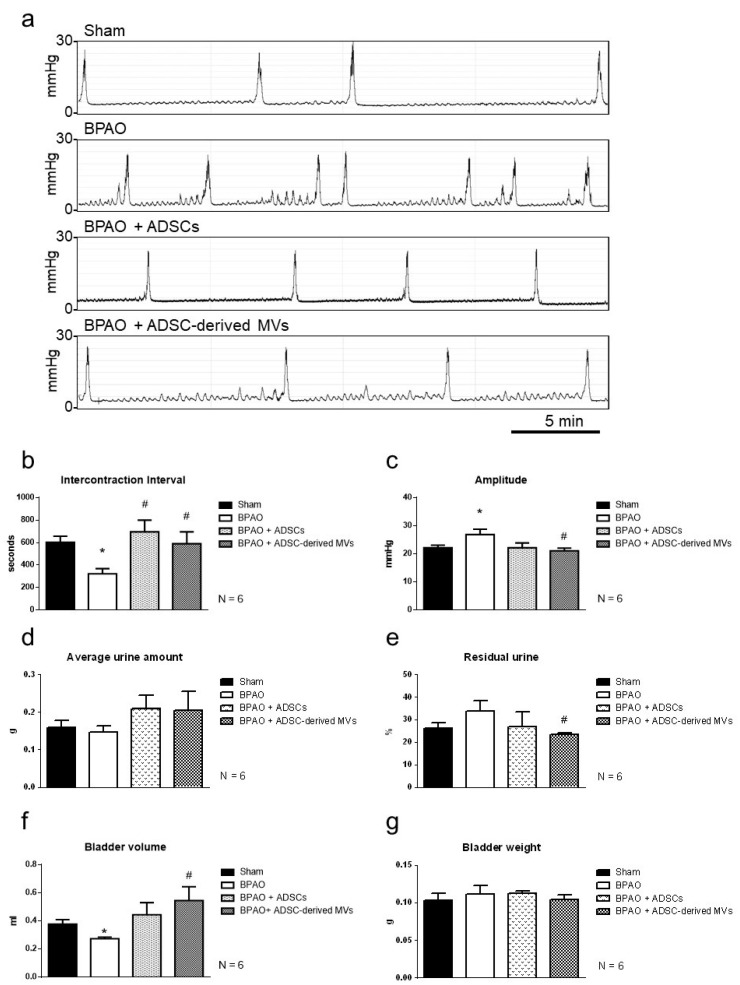
IVP patterns on transcystogram study in separated group. In the top panel, (**a**) demonstrated the IVP patterns on transcystometrogram. In the lower-left panel, (**b**) showed significantly lower ICI in BPAO group than sham group (* *p* < 0.05). The ICI in BPAO + ADSCs group and BPAO + ADSC-derived MVs group is higher than BPAO group (^#^ *p* < 0.05). In the lower-right panel, (**c**) showed that the amplitude level is significantly higher in BPAO group than sham group (* *p* < 0.05). It also showed that the amplitude level is significantly lower in BPAO + ADSC-derived MVs group compared to BPAO group (^#^ *p* < 0.05). (**d**) showed that the average urine amount was of no significant differences among the four groups. (**e**) showed that the residual urine amount is significantly lower in BPAO + ADSC-derived MVs group than BPAO group (^#^ *p* < 0.05). In the bottom-left panel, (**f**) showed that the bladder volume is significantly lower in BPAO group than sham group (* *p* < 0.05). It also showed that the BPAO + ADSC-derived MVs group had higher bladder volume than BPAO group (^#^ *p* < 0.05). In the bottom-right panel, (**g**) showed that the bladder weight is of no significant differences between groups. The detailed data are shown in Table 2.

**Figure 5 ijms-22-07000-f005:**
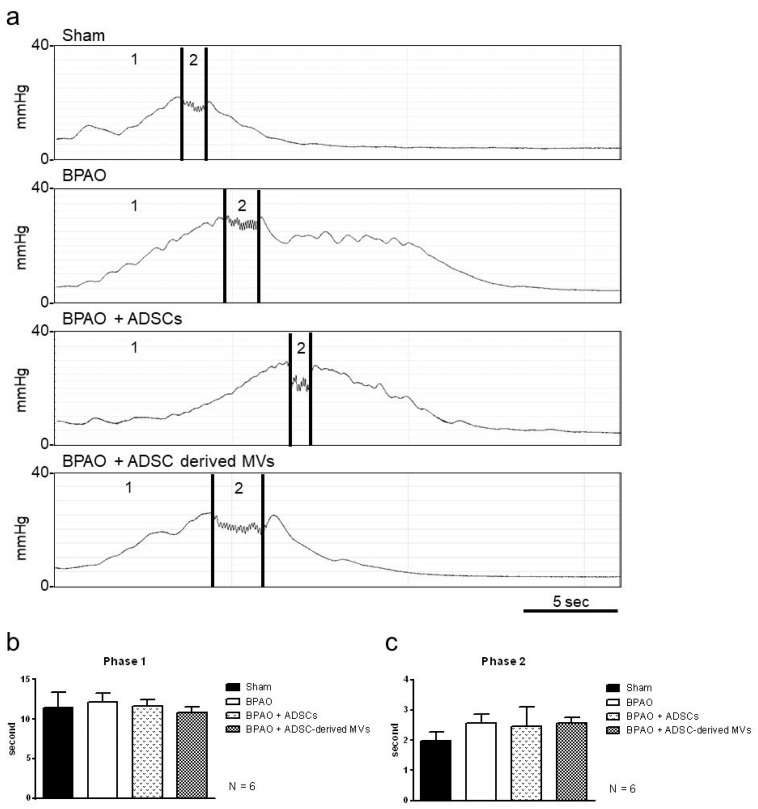
Micturition contraction patterns on transcystogram study in separated group. In the top panel, (**a**) demonstrated the micturition contraction patterns on transcystometrogram. In the bottom panel, (**b**,**c**) showed that the duration of phasic 1 contraction and phasic 2 HFO were of no significant differences between groups. The detailed data are shown in Table 3.

**Figure 6 ijms-22-07000-f006:**
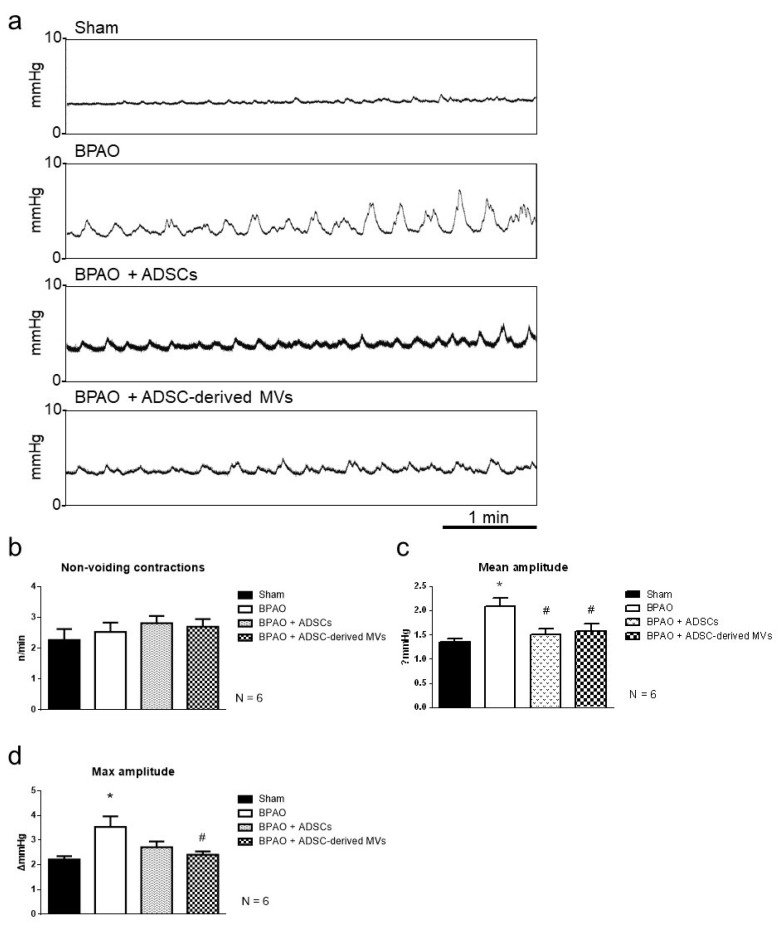
NVC patterns on transcystogram study in separated group. In the top panel, (**a**) demonstrated the NVC on transcystometrogram. In the middle-left panel, (**b**) showed the number of NVC among groups. The number of NVC is of no significant differences between groups. In the middle-right panel, (**c**) showed the mean amplitude of NVC. The mean amplitude of NVC is significantly higher in BPAO group than sham group (* *p* < 0.05). The mean amplitude of NVC is significantly lower in BAPO + ADSCs and BPAO + ADSC-derived MVs group than BPAO group (^#^ *p* < 0.05). In the bottom-left panel, (**d**) showed the maximum amplitude of NVC. The maximum amplitude of NVC is significantly higher in BPAO group than sham group (* *p* < 0.05). The maximum amplitude of NVC is significantly lower in BPAO + ADSC-derived MVs group than BPAO group (^#^ *p* < 0.05). The detailed data are shown in Table 4.

**Figure 7 ijms-22-07000-f007:**
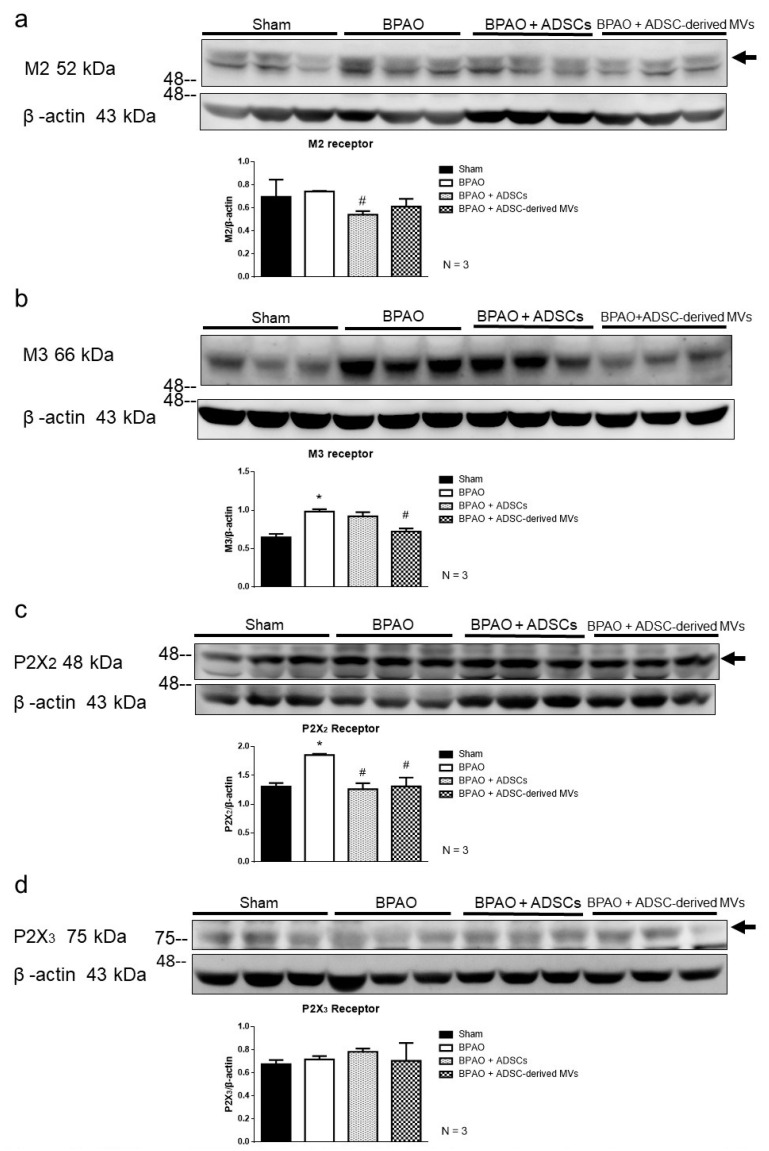
Purinergic and muscarinic signaling associated molecules changes on western blot study in separated group. Figure 7 showed the expression of muscarinic cholinergic (M2 and M3) and purinergic receptor proteins (P2X2 and P2X3) by Western blot. In the top panel, (**a**) showed the expression of M2 receptor is significantly decreased in BPAO + ADSCs group compared to BAPO group (^#^ *p* < 0.05). The expression of M2 receptor had trend of decreasing in BPAO + ADSC-derived MVs group compared to BAPO group without significant differences (*p* > 0.05). In the middle panel, (**b**) showed that the expression of M3 receptor was significantly increased in BPAO group compared to sham group (* *p* < 0.05). The expression of M3 receptor is significantly decreased in BPAO + ADSC-derived MVs group compared to BAPO group (^#^ *p* < 0.05). (**c**) showed the expression of P2X2 receptors was significantly increased in BPAO group compared to sham group (* *p* < 0.05). The expression of P2X2 receptor is significantly decreased in BPAO + ADSC-derived MVs and BAPO + MVs group compared to BAPO group (^#^*p* < 0.05). In the bottom panel, (**d**) showed the expression of P2X3 receptor is of no significant differences between groups.

**Figure 8 ijms-22-07000-f008:**
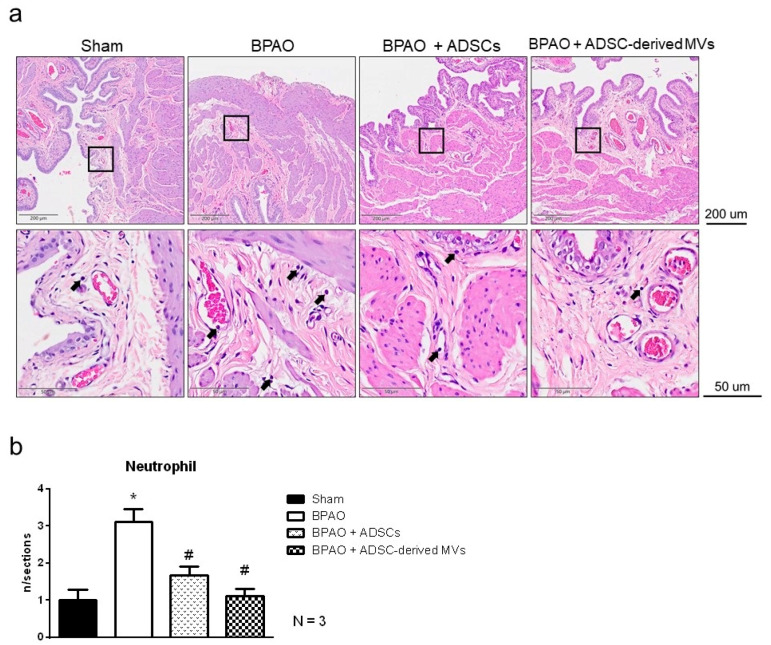
Histological staining and analyses in separated group. In the top panel, (**a**) showed the H&E staining among groups. In the bottom panel, (**b**) showed analysis of neutrophil counts. The results showed BPAO group have the highest number of neutrophil infiltration among the four groups (* *p* < 0.05). The neutrophil counts of BPAO + ADSCs groups and BPAO + ADSC-derived MVs group are significantly less than BPAO group (^#^ *p* < 0.05).

**Figure 9 ijms-22-07000-f009:**
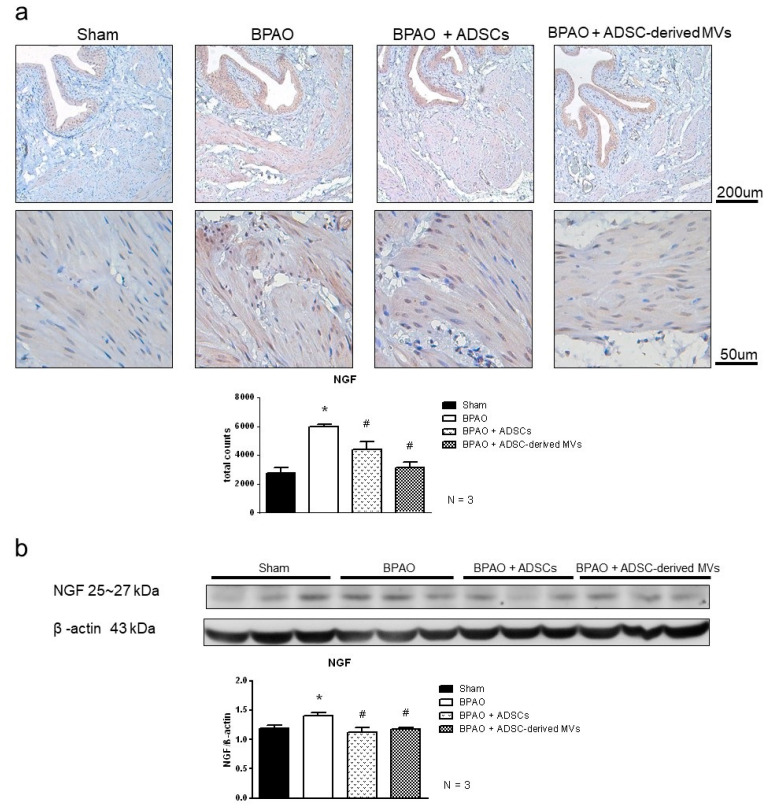
NGF staining and expression in separated group. In the top panel, (**a**) showed the NGF staining and image analysis of NGF staining. It showed significantly increased NGF density in BPAO group compared to sham group. The NGF density in BPAO + ADSCs and BPAO + ADSC-derived MVs group were significantly decreased compared to BPAO group (^#^ *p* < 0.05). In the bottom panel, (**b**) showed the western blot of NGF level. It showed significantly increased level of NGF expression in BPAO group compared to the other groups (* *p* < 0.05). The NGF level is lower in BPAO + ADSCs group and BPAO + ADSC-derived MVs group compared to BPAO group (^#^ *p* < 0.05).

**Figure 10 ijms-22-07000-f010:**
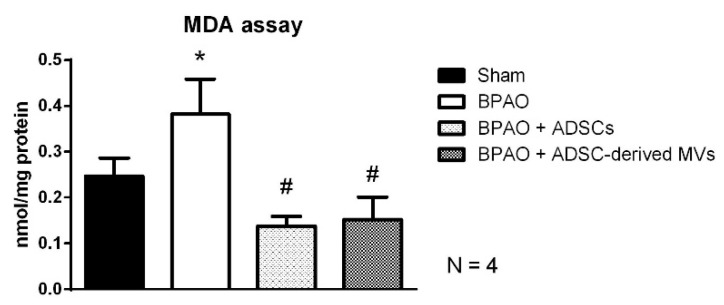
MDA assay in separated group. Figure 10 showed BPAO group had significantly higher level of MDA than sham group (* *p* < 0.05). The MDA level is lower in BPAO + ADSCs group and BPAO + ADSC-derived MVs group compared to BPAO group (^#^ *p* < 0.05).

**Table 1 ijms-22-07000-t001:** Detailed data of behavior parameters among groups on metabolic cage study. Data are presented as mean ± standard deviation; the 95% confidence intervals are shown in brackets. * *p* < 0.05 vs. sham group by two-tailed non-paired Student’s *t*-test. ^#^ *p* < 0.05 vs. BPAO group by two-tailed non-paired Student’s *t*-test.

Group	Urination Frequency (n/Half Day)	Water Intake (mL/Day)	Food Intake (g/Day)	Urine (g/Day)	Feces (g/Day)
Sham, n = 6	7.5 ± 1.67 [5.65, 9.35]	38 ± 13.49 [23.84, 52.16]	17.7 ± 8.67 [8.61, 26.79]	19.33 ± 2.8 [16.39, 22.27]	14.03 ± 3.35 [10.51, 17.56]
BPAO, n = 6	10.83 ± 1.47 * [9.29, 12.38]	47.83 ± 13.04 [34.14, 61.52]	21.43 ± 3.09 [18.19, 24.68]	22.88 ± 5.84 [17.75, 30.01]	18.5 ± 5.99 [21.21, 24.79]
BPAO + ADSCs, n = 6	7.68 ± 1.37 ^#^ [6.23, 9.1]	40.17 ± 15.94 [23.44, 56.9]	15.27 ± 7.99 [6.92, 23.62]	19.03 ± 5.92 [13.09, 25.51]	15.02 ± 4.79 [9.99, 20.04]
BPAO + ADSC-derived MVs, n = 6	7.83 ± 1.472 ^#^ [6.29, 9.38]	45.83 ± 3.13 [42.55, 49.11]	21.3 ± 1.64 [19.58, 23.03]	19.50 ± 6.68 [12.49, 26.51]	13.92 ± 2.77 [11.01, 16.83]

**Table 2 ijms-22-07000-t002:** Detailed data of transcystometrogram. Data are presented as mean ± standard deviation; the 95% confidence intervals are shown in brackets. * *p* < 0.05 vs. sham group by two-tailed non-paired Student’s *t*-test. ^#^ *p* < 0.05 vs. BPAO group by two-tailed non-paired Student’s *t*-test.

Group	Intercontraction Interval (s)	Amplitude (mmHg)	Average Urine Amount (g)	Residual Urine (%)	Bladder Volume (mL)	Bladder Weight (g)
Sham, n = 6	602.8 ± 132.3 [463.9, 741.7]	22.44 ± 2.64 [19.67, 25.21]	0.16 ± 0.043 [0.12, 0.21]	26.12 ± 6.11 [19.70, 32.53]	0.38 ± 0.08 [0.29, 0.46]	0.10 ± 0.02 [0.08, 0.13]
BPAO, n = 6	353.2 ± 149.0 * [196.8, 509.5]	26.45 ± 5.11 * [21.09, 31.82]	0.15 ± 0.038 [0.11, 0.19]	33.97 ± 10.68 [22.76, 45.17]	0.27 ± 0.028 * [0.24, 0.30]	0.11 ± 0.022 [0.08, 0.15]
BPAO + ADSCs, n = 6	592.0 ± 293.8 ^#^ [283.6, 900.3]	22.16 ± 4.21 [17.74, 26.57]	0.21 ± 0.088 [0.12, 0.30]	26.88 ± 16.49 [9.57, 44.19]	0.44 ± 0.21 [0.22, 0.67]	0.11 ± 0.01 [0.10, 0.12]
BPAO + ADSC-derived MVs, n = 6	575.6 ± 269.8 ^#^ [292.4, 858.8]	21.10 ± 2.33 ^#^ [18.65, 23.55]	0.21 ± 0.12 [0.07, 0.34]	23.52 ± 14.79 ^#^ [21.97, 25.08]	0.71 ± 0.37 ^#^ [0.31, 1.01]	0.10 ± 0.01 [0.09, 0.12]

**Table 3 ijms-22-07000-t003:** The duration of phasic 1 and phasic 2 micturition contraction on transcystometrogram. Data are presented as mean ± standard deviation; the 95% confidence intervals are shown in brackets. * *p* < 0.05 vs. sham group by two-tailed non-paired Student’s *t*-test. ^#^ *p* < 0.05 vs. BPAO group by two-tailed non-paired Student’s *t*-test.

Group	Phase 1	Phase 2
Sham, n = 6	11.43 ± 4.91 [6.28, 16.58]	1.98 ± 0.72 [1.22, 2.74]
BPAO, n = 6	12.21 ± 2.58 [9.50, 14.92]	2.57 ± 0.73 [1.81, 3.33]
BPAO + ADSCs, n = 6	11.64 ± 2.17 [9.36, 13.92]	2.46 ± 1.59 [0.79, 4.13]
BPAO + ADSC-derived MVs, n = 6	10.81 ± 1.88 [8.84, 12.79]	2.58 ± 0.46 [2.1, 3.05]

**Table 4 ijms-22-07000-t004:** Analysis of NVC on transcystometrogram. Data are presented as mean ± standard deviation; the 95% confidence intervals are shown in brackets. * *p* < 0.05 vs. sham group by two-tailed non-paired Student’s *t*-test. ^#^ *p* < 0.05 vs. BPAO group by two-tailed non-paired Student’s *t*-test.

Group	Non-Voiding Contraction (n)	Mean Amplitude (ΔmmHg)	Max Amplitude (ΔmmHg)
Sham, n = 6	2.27 ± 0.87 [1.35, 3.18]	1.35 ± 0.18 [1.15, 1.54]	2.22 ± 0.34 [1.86, 2.57]
BPAO, n = 6	2.54 ± 0.74 [1.76, 3.31]	2.09 ± 0.42 * [1.65, 2.53]	3.54 ± 1.05 * [2.44, 4.64]
BPAO + ADSCs, n = 6	2.82 ± 0.59 [2.20, 3.43]	1.50 ± 0.32 ^#^ [1.16, 1.84]	2.71 ± 0.59 [2.10, 3.33]
BPAO + ADSC-derived MVs, n = 6	2.71 ± 0.60 [2.08, 3.32]	1.58 ± 0.37 ^#^ [1.19, 1.97]	2.41 ± 0.32 ^#^ [2.07, 2.75]

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
