# Peer review of "Adipose-Derived Stem Cells and Their Derived Microvesicles Ameliorate Detrusor Overactivity Secondary to Bilateral Partial Iliac Arterial Occlusion-Induced Bladder Ischemia"

_ijms, 2021, doi:10.3390/ijms22137000_

Round 1

Reviewer 1 Report

Dear Authors,

I have read your manuscript with interest and I have very good opinion about it. I would only reccommend to change a little an abstract - as I think it is not necessary to underline the parts of the abstract.

Author Response

Point 1: I have read your manuscript with interest and I have very good opinion about it. I would only reccommend to change a little an abstract - as I think it is not necessary to underline the parts of the abstract.

Response 1: Thanks for your comments. We have removed the underline in the abstract.

Reviewer 2 Report

General Critique of Work

1/ Author names are spelled out and institutional affiliations are signified with footnotes. Likewise, corresponding author is noted with an asterisk in the author list.

2/ Numbers (a, b, c…) associated to figures are not really clear, authors have to address them in a better position, in the top-right for example

2/ Line 78, there are not only microRNA, authors might remove microRNA or add protein.

3/ Figure 1b: p-value obtained from statistical test to analyze the average blood flow remain surprisingly, another test will be more judicious such t test by separating left and right arteries.

3/ This study was performed by using 6 rats by condition, i.e 24 animals, authors have performed a duplicates of their experiments?

5/ Do Microvesicles derive from the same ADSC cells supernatant, or authors have performed 6 independent cultures of ADSC? This poins is really not clear making conclusion of the article really confusing.

6/ In a general manner, animals have received pain-killer following bladder ischemia?

7/ Authors have to show microvesicles markers justifying their purification as well as the number (concentration) or the amount of microvesicles injected.

8/ It’s not clear concerning ADSC injection whether cells colonize ischemia region. Do Authors have an idea where ADSC cells spread?

Conclusion:

Authors have performed a good job, however it’s not clear whether authors have duplicate their experiment on another cohort of rats and whether microvesicles remain purified from independent ADSC cultures. Authors have to improve their experimental approaches concerning microvesicles by taking in consideration the above remarks and perform supplemental experiments. Then, this article will be suitable for publication in Cells.

Round 2

Reviewer 2 Report

I thank authors for taking my comments into account.
I believe that the manuscript is now suitable for publication in IJMS.